# Cadmium Is Associated with Type 2 Diabetes in a Superfund Site Lead Smelter Community in Dallas, Texas

**DOI:** 10.3390/ijerph17124558

**Published:** 2020-06-24

**Authors:** Bert B. Little, Robert Reilly, Brad Walsh, Giang T. Vu

**Affiliations:** 1School of Public Health and Information Sciences, University of Louisville, Louisville, KY 40292, USA; rreilly@uabmc.edu (R.R.); giang.vu@louisville.edu (G.T.V.); 2Department of Anthropology, University of Louisville, Louisville, KY 40292, USA; 3Parkland Hospital and Health System, Dallas, TX 75235, USA; D.WALSH@phhs.org; 4Nephrology Section, Birmingham School of Medicine, University of Alabama, Birmingham, AL 35233, USA

**Keywords:** cadmium, Type 2 diabetes, EPA superfund, African-American

## Abstract

Objective: To test the hypothesis that cadmium (Cd) exposure is associated with type 2 diabetes mellitus (T2DM). Materials and Methods: A two-phase health screening (physical examination and laboratory tests) was conducted in a lead smelter community following a Superfund Cleanup. Participants were African Americans aged >19 years to <89 years. Multiple logistic regression was used to analyze T2DM regressed on blood Cd level and covariates: body mass index (BMI), heavy metals (Ar, Cd, Hg, Pb), duration of residence, age, smoking status, and sex. Results: Of 875 subjects environmentally exposed to Cd, 55 were occupationally exposed to by-products of lead smelting and 820 were community residents. In addition, 109 T2DM individuals lived in the community for an average of 21.0 years, and 766 non-T2DM individuals for 19.0 years. T2DM individuals (70.3%) were >50 years old. Blood Cd levels were higher among T2DM subjects (*p* < 0.006) compared to non-T2DM individuals. Logistic regression of T2DM status identified significant predictors: Cd level (OR = 1.85; 95% CI: 1.14–2.99, *p* < 0.01), age >50 years (OR = 3.10; 95% CI: 1.91–5.02, *p* < 0.0001), and BMI (OR = 1.07; CI: 1.04–1.09, 0.0001). In meta-analysis of 12 prior studies and this one, T2DM risk was OR = 1.09 (95% CI: 1.03–1.15, *p* < 0.004) fixed effects and 1.22 (95% CI: 1.04–1.44, *p* < 0.02) random effects. Discussion: Chronic environmental Cd exposure was associated with T2DM in a smelter community, controlling for covariates. T2DM onset <50 years was significantly associated with Cd exposure, but >50 years was not. Meta-analysis suggests that Cd exposure is associated with a small, but significant increased risk for T2DM. Available data suggest Cd exposure is associated with an increased propensity to increased insulin resistance.

## 1. Introduction

Cadmium (Cd) is a trace element, occurring in about 0.1 to 0.5 ppm in the earth’s crust [1]. It is chemically similar to zinc, but, unlike zinc, Cd is not part of normal animal metabolic processes. Its physicochemical properties allow it to assume positions in metabolic pathways that zinc and copper normally occupy [2,3]. Cd is toxic to higher organisms because it disrupts normal metabolism and accumulates in organs. Cd accumulates in kidney in humans, and is particularly nephrotoxic because it concentrates in proximal tubular cells [4]. Secondarily, Cd accumulates in lungs, bone, and liver [5]. In addition, Cd is associated with type 2 diabetes mellitus (T2DM) onset, cardiovascular disease, and osteoporosis [2,6].

Cd is toxic to humans, and was first recognized in 1858 [7]. Uptake through the gastrointestinal route is small, approximately 5% of ingested Cd, although Cd can accumulate in vegetable foods. Human lung resorbs 40–60% of Cd in tobacco smoke, which is the major source of Cd inhalation [1]. Once absorbed, the majority of Cd circulating in blood is bound to proteins (e.g., albumin, metallothionein). Liver is the first organ to process absorbed Cd, and where Cd induces the production of Cd-metallothionein [7]. Kidney damage associated with Cd exposure is a well-recognized problem for exposed patients [2]. Cd reaches the kidney in the form of cadmium-metallothionein (Cd-MT). Cd-MT is filtered by the glomerulus, and subsequently reabsorbed in proximal tubule, where it remains, comprising the major part of Cd body burden. Cd concentration in proximal tubular cells increases over a person′s life span, with toxicity expressed as chronic kidney disease. In this study, the frequency of chronic kidney disease stage (CKD3) was increased in frequency among those exposed to Cd [8].

Cd exposure in humans is through inhalational (tobacco smoking) and oral (food) routes, but absorption from inhalation accounts for the greatest accumulation. Cd accumulates in the mass of most plants, particularly tobacco plants. Smokers consequently have kidney Cd concentrations 4 to 5-fold greater than the general non-smoking population, and these concentrations persist for decades. For non-smokers, the main source of Cd exposure is food consumed. Cd is concentrated in cereals (i.e., cereal products), vegetables, nuts and pulses, starchy roots, or potatoes. In turn, those animals fed these products subsequently concentrate Cd in meat and meat products. Cd exposure also occurs environmentally (e.g., dust, airborne particulates) from smelting activities (e.g., lead, zinc, copper).

T2DM is associated with Cd exposure as measured in blood/urine levels, often from exposure to by-products of lead smelting. Published literature has reported that Cd exposure causes increased insulin resistance [9]. Increased insulin resistance ultimately increases the propensity to develop T2DM. Cd increases pro-inflammatory action and monocyte chemoattractant protein-1 (MCP-1) expression is upregulated. Adipose tissue Cd accumulation leads to increased endocrine dysfunction and increased lipolysis. Adipocyte glucose dysregulation contributes to hyperglycemia. These and other influences increase the propensity to have insulin resistance [9]. An estimated 12 studies have analyzed the association [10,11,12,13,14,15,16,17,18,19,20,21] and half of them [14,15,16,18,20] found a significant association of Cd with T2DM. Analyses of large populations prospectively are needed to more thoroughly evaluate the association between Cd exposure and T2DM. No clear consensus for the association was found across ten investigations [10,11,12,13,15,16,17,18,19,20].

The purpose of our investigation is to analyze the association between T2DM and exposure to cadmium and several other heavy metals, and conduct a meta-analysis across published studies.

## 2. Materials and Methods

The present investigation was approved by the Institutional Review Board at University of Texas Southwestern Medical Center, Dallas, Texas (IRB#0902-495, 4 September 2002), that includes adherence to the declaration of Helsinki.

### 2.1. Study Population

In 1934, the first of three lead smelters was established in West Dallas (RSR), at the intersection of Singleton Boulevard and Westmoreland Street. Two additional smelters were opened, across the street from each other in East Dallas, in the part of the city known as Cadillac Heights in the 1940s. The city of Dallas initiated health surveys and established lead clinics to provide testing after recognizing that environmental lead exposure was a public health problem in the 1960s. Several clean-up projects that included soil removal were conducted over the years. Ultimately, blood lead levels decreased only when the smelters closed. The companies were smelting lead from batteries and slag, releasing dust clouds whose plumes were blown by prevailing winds over housing in these communities [22]. One of Dallas′ largest public housing projects has grown up around the West Dallas smelter since 1934. The last smelter closed in 1990, and the West Dallas area was designated an Environmental Protection Agency (EPA) Superfund Site. Top soil was again removed and old housing demolished or renovated following the EPA cleanup. New public housing was constructed within the old smelter community site.

### 2.2. Study Design

In 2002, a two-phase health screening was conducted in Dallas lead smelter communities following an EPA Superfund Cleanup. Phase I was a demographic and risk survey. The sample is not random. It is a ‘sample of convenience,’ as frequently used in health surveys. Of the 12 previously published studies used for comparison, only four were random samples. Five were samples of convenience, two were NHANES secondary analyses (representative probability based cluster samples), and one was all workers in four coke factories in Wuhan, China. The samples were all comprised mainly of working and lower income (SES-socioeconomic status) individuals based on the monthly wage, very similar to the study sample which lived in middle and low income public housing. Participants were recruited through town hall meetings and public service announcements in the media (newspapers, radio, and television). People who lived in the following zip codes were included: 75208, 75211, 75212, 75247, 75203, 75215, and 75216. The vast majority were from the West Dallas smelter community. Self-selected participants were invited to return for Phase II, which included a physical examination and clinical laboratory tests. Subjects were required to have valid values listed for age, sex, race, creatinine, BMI, blood pressure, heavy metal (Ar, Cd, Hg, Pb) concentrations, GGT, HbA1c, lived in a smelter area zip code, and tobacco smoking status to be included in analyses. BMI was used as a continuous variable because each point change in the BMI is meaningful at the high end of the BMI distribution, and it has a standard error of 1. We tested using categories of overweight and obesity, and found that all subjects were obese (BMI > 30). Subjects with hypertension (blood pressure ≥ 140/90 mmHg) were dummy coded (0, 1) for regression analysis. Smoking was defined as current smoker (1/2 pack of cigarettes or more per day = 1) or non-smoker (0). Seven hundred and seventy-five subjects met these criteria. Of these, 55 males were smelter-working residents and 820 were non-smelter working community residents. Fourteen female smelter-working residents were excluded from the analysis because of missing data.

### 2.3. Type 2 Diabetes Mellitus Status

HbA1c was analyzed in study participants. The cut-point for T2DM was 6.5%, as recommended by the American Diabetes Association [23]. Subjects with diabetes mellitus (HbA1c ≥ 6.5%) were dummy coded (0, 1) for regression analysis.

### 2.4. Analytical Techniques

Two analytical approaches were used in the present study. We used logistic regression to analyze the ability of demographics and heavy metals to predict T2DM status. An analysis of the published literature was done using a systematic literature search and meta-analysis of studies that met inclusion criteria (reported primary data, sample sizes, Cd levels, method of Cd level determination).

Multivariate logistic regression was used to analyze the dependence of T2DM (0, 1) on independent variables. Our null hypothesis tested was that T2DM is not dependent on exposure to Cd or other metals. The regression model was:

T2DM (0, 1) = age GT 50 years (0, 1) + gender: male (0, 1) + smoking tobacco (0, 1) + duration of residence (decades) + smelter worker (0, 1) + log_10_ (blood lead level) + log_10_ (blood arsenic level) + log_10_ (blood mercury level) + log_10_ (blood cadmium level) + gamma-glutamyl-transpeptidase (GGT) abnormal (0, 1) + hypertension (0, 1), where 0 = No, 1 = Yes.

Multicollinearity was assessed with multiple regression variance inflation factor (VIF) using the model above. VIFs were generally less than 1.45. The VIFs for demographics (age—1.41, gender—1.27, smelter worker—1.19, duration—1.10, smoker—1.09), physical characteristics (BMI—1.17, hypertension—1.27, GGT—1.13), and heavy metals (Ar—1.08, Cd—1.17, Hg—1.08, Pb—1.29) are low. Thus, VIFs indicate multicollinearity is likely not confounding these results.

Meta-analysis was done to obtain estimated composite ORs and 95% Confidence Intervals (CI) under fixed and random effects, and measures of heterogeneity. The heterogeneity was assessed using the Q and I^2^ statistics. A PRISMA compliant literature review was done. Using pubmed.gov and scholar.google.com for years 1960 to 31 December 2019, search terms were (Cadmium AND Diabetes), which resulted in 372 studies. A subset of those studies was searched using (Human AND Type 2 Diabetes) to refine the search to humans with T2DM, with 32 studies meeting those search criteria. Manual review excluded studies were those that did not report primary data (described below), or did not report Cd levels or method of testing. The presence of primary data on study parameters was required for inclusion of studies, as described below, resulted in 12 studies of Cd and T2DM for comparison with our results, excluding studies that reported only secondary data. Studies chosen for inclusion had to present primary data analysis on the association of Cd with T2DM. Investigations included were limited to those that reported sample sizes of Cd exposed and controls, and those that reported T2DM status. In addition, the chemical method of determination had to be included, and the biological fluid used for analysis (blood, urine), and T2DM diagnostic criteria. Software SAS (v.9.4, SAS Institute, Cary, NC, USA) and SPSS (v.25, IBM, Chicago, IL, USA) were used to analyze the data. Before the log transformation of heavy metal concentration was done, 1.0 was added to allow calculation of a log for a concentration of zero, which resulted in those with no exposure having a score of zero (i.e., log 1 = 0). Therefore, 1.0 for the metals (Ar, Cd, Hg, Pb) represents a concentration of 10 µg/L). Meta-analysis was done using Comprehensive Meta-Analysis, Version 3 (Biostat, Englewood, NJ, USA).

## 3. Results

The prevalence of T2DM in the study group was 14.2% (95% CI: 12.2–16.2%). Univariate comparison of variables by T2DM status revealed that age was markedly different between T2DM and non-T2DM groups by one-way ANOVA, 55.0 years vs. 45.7 years, respectively (*p* < 0.001). Therefore, study variables are broken down by age < and age >50 years (Table 1).

### 3.1. Total Sample

Cd levels differed significantly between T2DM vs. non-T2DM (*p* < 0.006), but the other heavy metals (arsenic, lead, mercury) did not differ. BMI was markedly higher among those with T2DM (*p* < 0.0001). Duration of residence was slightly different with T2DM participants having lived in the smelter community an average of 21 years compared to non-T2DM participants who had resided there for a mean of 19 years (*p* < 0.11).

There was an excess of females in the T2DM group by 6% (*p* < 0.07), and 31.6% more of those aged >50 years in the diabetes group, which is similar to other studies of T2DM [17,20]. In addition, CKD3 was significantly (*p* < 0.0001) more frequent in the T2DM group compared to the non-T2DM group. Smoking tobacco and abnormal GGT did not differ significantly between the two groups. T2DM was more frequent among smelter workers (*n* = 43, 78.2%) compared to individuals who were not smelter workers (*p* < 0.03).

### 3.2. <50 Years Old

Among those <50 years old, T2DM individuals were significantly older by 4.4 years (*p* < 0.001). Mean Cd level was three-fold higher in the T2DM group than in the non-T2DM group (*p* < 0.02). Notably, duration of residence was 6.7 years longer among the T2DM individuals than among non-T2DM participants (*p* < 0.001). BMI was significantly higher among T2DM participants compared to non-T2DM individuals (42.5 v. 31.1, *p* < 0.0001).

There were 6.5% more females in the T2DM group (*p* < 0.07), and no other categorical variables approached significance. Notably, CKD Stage 3 was 1.1% higher among non-T2DM individuals (*p* < 0.12).

### 3.3. >50 Years Old

In the older age group, average age was not different between T2DM and non-T2DM sub-groups. BMI was higher (*p* < 0.0001) among T2DM individuals (34.4) compared to non-T2DM participants (32.8). The difference between the two groups in frequency of CKD Stage 3, with 8.7% more cases in the T2DM group was also significant (*p* < 0.04).

Logistic regression was used to further analyze the association of T2DM with Cd blood levels controlling for covariates.

### 3.4. Total Sample

The odds ratio (OR) for Cd was 1.85 (95% CI: 1.14–2.99, *p* < 0.01). Age was highly significant in predicting T2DM with those 50 years and older being 3.31 times more likely to have T2DM (*p* < 0.0001). BMI was also highly significant with each unit increase in the measure associated with a 7% increase in odds of having T2DM (*p* < 0.0001). Other independent variables were not significant in the fully adjusted model (Table 2).

### 3.5. <50 Years Old

In the younger age group, four variables significantly predict T2DM. Cd blood level had an OR of 2.69 (95% CI: 1.23–5.90, *p* < 0.01). For each unit increase in BMI, the odds of having T2DM increased 10% (*p* < 0.0001). Duration of residence in the community significantly increased the odds of having T2DM, OR = 1.50 (95% CI: 1.10–2.05, *p* < 0.01). Hypertension had a large effect increasing the risk for T2DM with OR = 4.39 (*p* < 0.007). Importantly, Cd was significant in <50-year-olds but not in >50 year olds (Table 2). Cd exposure is significantly associated with T2DM onset in those <50 years, but not among individuals >50 years.

### 3.6. >50 Years Old

Among older participants, Cd blood level is not significant (OR = 1.59, 95%CI: 0.83–3.05, *p* < 0.17). BMI was a significant but small risk factor for T2DM in the >50 year old sub-group. Hypertension significantly increased the risk of T2DM (OR = 2.80, *p* < 0.03). No other covariates were significant (Table 2).

### 3.7. Meta-Analysis of Cadmium and T2DM

Five published investigations of the association between T2DM and blood Cd, and eight that used urinary Cd met inclusion criteria for meta-analysis. Meta-analysis of blood Cd levels association with T2DM resulted in fixed effects OR of 1.19 (95% CI: 1.03–1.39, *p* < 0.02), and a random effects OR of 1.26 (95% CI: 0.92–1.71) (Table 3). Fixed effects model yielded an OR of 1.19 (95% CI: 1.03–1.39, *p* < 0.02). Significant heterogeneity in the analysis (Q = 31.19, 0.0001; I^2^ = 74.35) was found, as apparent in the forest plot (Table 3).

Meta-analysis of urine Cd level and T2DM under the fixed effects model, the OR was 1.07 (95% CI: 1.01–1.13, *p* < 0.03) (Table 4). Under the random effects model, the OR of 1.20 (95% CI: 0.97–1.49, *p* < 0.09) was not significant, with moderately high heterogeneity, similar to blood Cd (Q = 31.19, 0.0001; I^2^ = 74.35). Meta-analysis of blood and urine Cd determinations combined, OR under the fixed effects model was 1.09 (95% CI: 1.03–1.15, *p* < 0.004), and under the random model OR was 1.22 (95% CI: 1.04–1.44, *p* < 0.02), and higher heterogeneity than either blood or urine separately (Q = 49.51, 0.0001; I^2^ = 74.35) (Table 5).

In the present investigation, Cd exposure significantly increased the odds of T2DM among those under 50 years (OR = 2.69, 95% CI: 1.23–5.90, *p* < 0.01) but not among those 50 years and older (OR = 1.59, 95% CI: 0.83–3.05, *p* < 0.17). Inclusion of this study’s findings resulted in significant meta-analysis results (OR = 1.09, *p* < 0.004, fixed effects, and OR = 1.22, *p* < 0.09, random effects).

## 4. Discussion

Cd exposure was significantly more frequent in T2DM in the age-combined sample with an OR = 1.85 (95% CI: 1.14–2.99, *p* < 0.01). Among adults younger than 50 years, the OR for Cd was 2.69 (95% CI: 1.23–5.90, *p* < 0.01), but was not significant among those ≥50 years (OR = 1.59, 95% CI: 0.83–3.05, *p* < 0.17). Lack of significance in the older age group may be associated with a Type II error because there are only 352 individuals in the age >50 year-old subsample.

Prior studies of the association of Cd exposure with T2DM are not consistent. One of five prior studies (20%) using blood Cd level found a significant association between the heavy metal and T2DM occurrence (Table 3). If the present study is included, two of six (33.3%) investigations found a significant association between blood Cd level and T2DM. Five of nine investigations (55.6%) that analyzed the association between urinary Cd levels and T2DM reported a significant association. A previously published meta-analysis did not detect an association between blood or urinary Cd and T2DM [24]. In further subgroup and sensitivity analyses, the association between Cd exposure and T2DM was not significant, and no publication bias was found.

OR magnitude and T2DM prevalence are associated, and may provide insights into understanding prior studies of diabetes and Cd exposure. Reassessment of the meta-analysis by simple regression of log_10_ OR on log_10_ T2DM rate shows that, in a population with a T2DM rate less than 10% (log10 = 1), no significant associations were found. It is possible that the statistical power of the T2DM signal associated with Cd exposure is not usually detectable in populations with a T2DM rate of less than 10% (Figure 1). Alternatively, if OR is approximately 3.0 or greater, T2DM rate can be lower than 10% and still have a sufficiently strong signal to detect a significant increase associated with Cd exposure. Therefore, the strength of the association seems related to the proportion of the population with T2DM. If the T2DM is <10%, it appears a greater proportion of the population must be affected by Cd, which influences the magnitude of the OR. The one exception to the apparent need to have a T2DM rate >10% is where Cd levels are high. The exception to this pattern is a study from Thailand in which the exposed group had an average urinary Cd of 8.6 µg/g (range: 6.4–10.7 µg/g), a relatively high level [18]. Hence, high concentrations may sufficiently increase signal strength, even when the rate of T2DM is lower than 10%, to a detectable level.

Meta-analysis of Cd and T2DM for urinary and blood combined resulted in ORs that were not different from the less heterogeneous individual groups that used only blood or urine, OR_Fixed_ = 1.09 (95% CI: 1.03–1.15, *p* < 0.004) or OR_Random_ = 1.22 (95% CI: 1.04–1.44, *p* < 0.02). In the meta-analysis of 12 prior studies and this one, T2DM risk was 1.09 (95% CI: 1.03–1.14, *p* < 0.004) in a fixed-effects model and 1.22 (95% CI: 1.04–1.27, *p* < 0.02) the random effects model, respectively, associated with a 10 µg/L increase in Cd level. All the estimated ORs in this study were within the 95% CI of other ORs.

### Possible Mechanism Pathway

Collectively, available evidence suggests an association of Cd exposure with increased insulin resistance through a complex system of relationships. Increased insulin resistance ultimately increases the propensity to develop T2DM. Cd alters transcription factors and increases pro-inflammatory signaling, decreasing adipocyte differentiation, and leading to adipose tissue dysfunction. Monocyte chemoattractant protein-1 (MCP-1) expression is upregulated by Cd exposure and a stimulated pro-inflammatory response, which increases Cd infiltration into adipose tissue (Figure 2). The response of adipose tissue to infiltration leads to adipose dysfunction, leads to increased endocrine dysfunction, decreased glucose transporter modulation and decreased insulin receptor activity, and increased lipolysis. Adiponectin levels are decreased, adipocyte glucose handling is altered, and free fatty acid levels are increased. Insulin resistance is increased through this network of influences, and insulin resistance in turn contributes to the association of Cd with T2DM [9]. The association of Cd exposure with T2DM is biologically plausible. The statistical analyses support the association. Analyses of large populations prospectively are needed to more thoroughly evaluate the association between Cd exposure and T2DM.

## 5. Conclusions

In conclusion, Cd apparently increases risk for T2DM, meta-analysis of ten studies combined supports the increased risk, and the association is biologically plausible. Collectively, available data support a small contribution of Cd to T2DM risk, between 9% and 22%. Prior studies that did not find a significant association of Cd exposure with had T2DM may have suffered from a Type II statistical error. Because of the known risks of cadmium intake involved with tobacco use, this study suggests that clinicians treating patients with a history of tobacco use be aware of the risks of kidney damage and T2DM that tobacco exposure or other cadmium ingestion (such as industrial or smelter exposures) presents.

## Figures and Tables

**Figure 1 ijerph-17-04558-f001:**
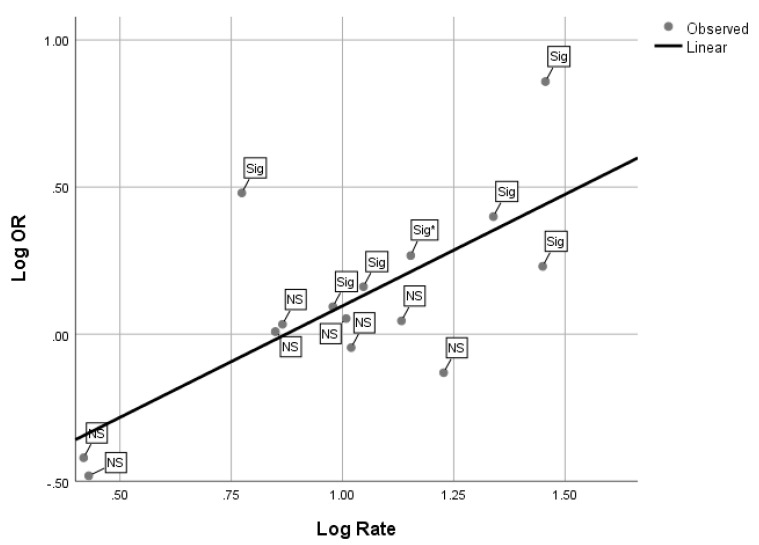
Analysis of log_10_ OR for regression of cadmium on T2DM status on log_10_ T2DM rate labeled by whether or not the OR was significant at *p* < 0.05 or lower. Regression statistics: R^2^ = 0.71, Adj. R2 = 0.47, B = 0.76 + 0.21 (*p* < 0.003), A = −0.66 + 0.22 (*p* < 0.01).

**Figure 2 ijerph-17-04558-f002:**
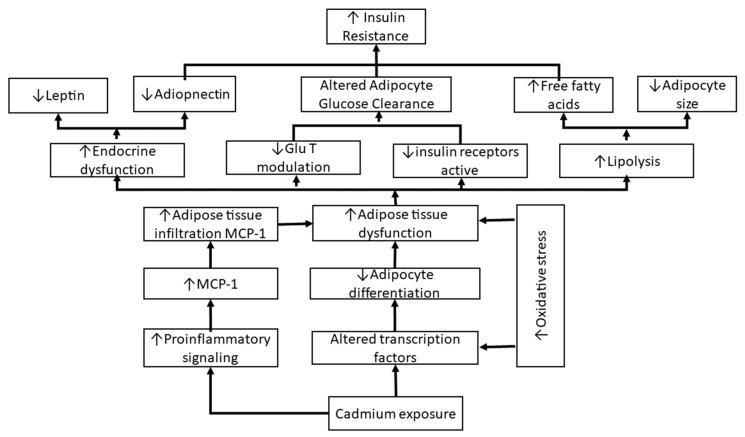
Possible pathway of relationship between Cd exposure and T2DM. (Modified from [9]).

**Table 1 ijerph-17-04558-t001:** Baseline characteristics.

**Total Sample (*n* = 875)**	**T2DM *** **(*n* = 109)** **95% CI**				**Non-T2DM** **(*n* = 766)** **95% CI**				***p***
	**M**	**SE**	**Lo**	**Hi**	**M**	**SE**	**Lo**	**Hi**	
Age	55.0	1.1	24.0	82.0	45.7	0.52	23.4	72.05	0.001
BMI (weight (kg)/height (m)^2^	37.0	1.0	35.0	38.9	31.9	0.3	31.3	32.4	0.0001
Arsenic (µg/L)	0.11	0.02	0.01	4.0	0.04	0.04	0.01	4.0	0.60
Cadmium (µg/L)	0.18	0.18	0.07	3.5	0.07	0.05	0.01	1.10	0.006
Lead (µg/dL)	2.38	0.25	1.00	23.0	2.25	0.08	0.00	33.0	0.48
Mercury (µg/L)	0.12	0.04	0.00	1.00	0.05	0.01	0.00	0.50	0.11
Duration of residence (years)	21.0	0.4	3.0	48.0	19.0	0.5	3.0	46.0	0.11
	***n***	**%**			***n***	**%**			
Females	72	66.1			462	60.3			0.07
Smoking	37	33.9			228	29.8			0.37
Age > 50 yrs.	74	67.9			278	36.3			0.0001
GGT abnormal	18	16.5			96	12.5			0.16
CKD Stage 3	16	14.7			35	4.6			0.0001
Smelter worker	43	78.2			12	21.8			0.03
**<50 Years old (*n* = 523)**	**T2DM *** **(*n* = 35)** **95% CI**				**Non-T2DM** **(*n* = 488)** **95% CI**				***p***
	**M**	**SE**	**Lo**	**Hi**	**M**	**SE**	**Lo**	**Hi**	
Age	41.6	1.3	24.0	49.0	36.9	0.4	19.0	49.0	0.001
BMI (weight (kg)/height (m)^2^	42.5	2.1	38.2	46.8	31.8	0.4	31.1	32.6	0.0001
Arsenic (µg/L)	0.11	0.11	0.00	4.00	0.12	0.03	0.01	4.0	0.96
Cadmium (µg/L)	0.21	0.11	0.00	3.50	0.07	0.01	0.01	4.5	0.02
Lead (µg/dL)	1.71	0.25	1.00	6.00	1.93	0.08	0.00	14.0	0.48
Mercury (µg/L)	0.12	0.04	0.00	1.00	0.05	0.02	0.00	7.0	0.10
Duration of residence (years)	24.0	0.4	1.0	49.0	17.3	0.5	1.0	48.0	0.001
	***n***	**%**			***n***	**%**			
Females	22	62.9			275	56.4			0.07
Smoking	37	33.9			228	29.8			0.28
GGT abnormal	7	20.0			60	12.3			0.15
CKD Stage 3	2	5.7			33	6.8			0.12
Smelter Worker	14	73.7			5	26.3			0.001
**>50 Years old (*n* = 352)**	**T2DM *** **(*n* = 74)** **95% CI**				**Non-T2DM** **(*n* = 278)** **95% CI**				***p***
	**M**	**SE**	**Lo**	**Hi**	**M**	**SE**	**Lo**	**Hi**	
Age	61.3	0.9	50.0	82.0	61.2	0.5	50	86.0	0.92
BMI (weight (kg)/height (m)^2^	34.4	0.89	32.6	36.1	31.9	0.5	31.0	32.8	0.02
Arsenic (µg/L)	0.00	0.00	0.00	0.00	0.09	0.03	0.00	4.0	0.20
Cadmium (µg/L)	0.17	0.06	0.00	2.80	0.09	0.02	0.00	3.0	0.11
Lead (µg/dL)	2.68	0.35	0.00	23.0	2.81	0.17	0.00	33.0	0.76
Mercury (µg/L)	0.10	0.05	0.00	2.00	0.06	0.02	0.00	3.0	0.41
Duration of residence (years)	19.6	1.6	3.0	71.0	22.1	0.9	1.0	64.0	0.19
	***n***	**%**			***n***	**%**			
Females	53	71.6			187	67.3			0.48
Smoking	24	67.6			75	27.0			0.35
GGT abnormal	11	14.9			36	12.9			0.67
CKD Stage 3	14	18.9			28	10.16			0.04
Smelter Worker	29	70.6			7	19.4			0.81

* HbA1c = T2DM when ≥6.5%, and T2DM = 1; population prevalence of T2DM = 12.5% (95% CI: 10.5–14.8%).

**Table 2 ijerph-17-04558-t002:** Binary logistic regression of T2DM on cadmium and covariates.

Variable	95% CI			
	OR	lo	hi	*p*
**Total Sample (*n* = 875)**				
Age > 50 years	3.10	1.91	5.02	0.0001
Arsenic (µg/L)	0.67	0.40	1.12	0.13
BMI wt_kg_/(ht_cm_)^2^	1.07	1.04	1.09	0.0001
Cadmium (µg/L)	1.85	1.14	2.99	0.01
Duration residence (decade)	1.00	0.86	1.17	0.997
Sex (F, M)	0.73	0.42	1.26	0.25
GGT abnormal (No, Yes)	1.38	0.73	2.61	0.33
Hypertension (No, Yes)	3.62	1.85	7.11	0.0001
Lead (µg/L)	0.99	0.89	1.10	0.83
Mercury (µg/L)	1.14	0.74	1.74	0.56
Smelter worker (No, Yes)	1.73	0.75	4.01	0.20
Smoking (No, Yes)	1.36	0.83	2.23	0.22
**<50 Years old (*n* = 523)**				
Arsenic (µg/L)	0.70	0.39	1.27	0.24
BMI wt_kg_/(ht_cm_)^2^	1.10	1.06	1.14	0.0001
Cadmium (µg/L)	2.69	1.23	5.90	0.01
Duration of residence (yrs.)	1.50	1.10	2.05	0.01
Sex (F, M)	0.83	0.32	2.16	0.70
GGT abnormal (No, Yes)	0.83	0.22	3.04	0.77
Hypertension (No, Yes)	4.39	1.51	12.74	0.007
Lead (µg/L)	0.81	0.62	1.07	0.15
Mercury (µg/L)	0.88	0.49	1.57	0.65
Smelter worker	8.96	1.78	45.08	0.008
Smoking (No, Yes)	1.89	0.79	4.52	0.15
**>50 Years old (*n* = 352)**				
Arsenic (µg/L)	0.01	0.00	102.79	0.999
BMI wt_kg_/(ht_cm_)^2^	1.04	1.004	1.07	0.03
Cadmium (µg/L)	1.59	0.83	3.05	0.17
Duration of residence (Dec.)	0.87	0.72	1.05	0.15
Sex (F, M)	0.74	0.37	1.48	0.39
GGT abnormal (No, Yes)	1.31	0.59	2.90	0.51
Hypertension (No, Yes)	2.80	1.14	6.89	0.03
Lead (µg/L)	1.02	0.91	1.15	0.71
Mercury (µg/L)	1.17	0.56	2.42	0.68
Smelter worker	0.98	0.35	2.73	0.97
Smoking (No, Yes)	1.31	0.71	2.42	0.39

**Table 3 ijerph-17-04558-t003:** Meta-analysis of blood cadmium and Type 2 diabetes ^§^.

**Study Name**		**Statistics for Each Study**						
		**Odds Ratio**		**Lower Limit**	**Upper Limit**	***z* Value**		***p* Value**
Moon, 2013 [10]		0.900		0.634	1.278	−0.589		0.556
Barregard, 2013 [11]		0.380		0.098	1.472	−1.401		0.161
Borne, 2014 [12]		1.110		0.821	1.501	0.677		0.498
Nie, 2016 [13]		1.130		0.877	1.456	0.946		0.344
Li, 2017 [14]		2.510		1.488	4.234	3.449		0.001
Present study		1.850		1.142	2.996	2.501		0.012
		1.194		1.026	1.391	2.289		0.022
**T2DM**	**Non-T2DM**		**95% CI**				**Statistic**	***p***
***n***	***n***	**Ages**	**OR**	**Lo**	**hi**	***p***		
1819	15,543	18–92						
*Effects Model*								
Fixed			1.19	1.03	1.39	0.02		
Random			1.26	0.92	1.71	0.15		
							Q = 31.19	0.0001
							I^2^ = 74.35	

^§^ Compiled in part from Wu et al. 2017.

**Table 4 ijerph-17-04558-t004:** Meta-analysis of urinary cadmium and Type 2 diabetes ^§^.

**Study Name**		**Statistics for Each Study**						
		**Odds Ratio**		**Lower Limit**	**Upper Limit**	***z* Value**	***p* Value**	
Barregard, 2013 [11]		0.330		0.099	1.099	−1.806	0.071	
Haswell-Elkins, 2017 [15]		7.220		1.549	33.646	2.518	0.012	
Son, 2015 [16]		1.700		1.059	2.728	2.199	0.028	
Menke, 2016 [17]		0.740		0.509	1.077	−1.573	0.116	
Tangvarasittichai, 2015 [18]		3.020		1.231	7.407	2.414	0.016	
Liu, 2016 [19]		1.080		0.619	1.886	0.271	0.787	
Swaddiwudipong, 2010 [21]		1.020		0.952	1.093	0.565	0.572	
Schwartz, 2003 A [20]		1.240		1.060	1.450	2.691	0.007	
Schwartz, 2003 B [20]		1.450		1.069	1.967	2.386	0.017	
		1.070		1.008	1.135	2.209	0.027	
**T2DM**	**Non-T2DM**		**95% CI**				**Statistic**	***p***
***n***	***n***	**Ages**	**OR**	**Lo**	**hi**	***p***		
2286	29,819	15–76						
*Effects Model*								
Fixed			1.07	1.01	1.13	0.03		
Random			1.20	0.97	1.49	0.093		
							Q = 31.19	0.0001
							I^2^ = 74.35	

^§^ Compiled in part from Wu et al. 2017. A, B—This study reported low (A) and high (B) cadmium exposure ORs separately.

**Table 5 ijerph-17-04558-t005:** Meta-analysis of blood or urinary cadmium and Type 2 diabetes ^§^.

**Study Name**	**Statistics for Each Study**							
	**Odds Ratio**		**Lower Limit**	**Upper Limit**	***z* Value**		***p* Value**	
Moon, 2013 [10]	0.900		0.634	1.278	−0.589		0.556	
Barregard, 2013 [11]	0.380		0.098	1.472	−1.401		0.161	
Borne, 2014 [12]	1.110		0.821	1.501	0.677		0.498	
Nie, 2016 [13]	1.130		0.877	1.456	0.946		0.344	
Li, 2017 [14]	2.510		1.488	4.234	3.449		0.001	
Present study	1.850		1.142	2.996	2.501		0.012	
Barregard, 2013 [11]	0.330		0.099	1.099	−1.806		0.071	
Haswell-Elkins, 2017 [15]	7.220		1.549	33.646	2.518		0.012	
Son, 2015 [16]	1.700		1.059	2.728	2.199		0.028	
Menke, 2016 [17]	0.740		0.509	1.077	−1.573		0.116	
Tangvarasittichai, 2015 [18]	3.020		1.231	7.407	2.414		0.016	
Liu, 2016 [19]	1.080		0.619	1.886	0.271		0.787	
Swaddiwudipong, 2010 [21]	1.020		0.952	1.093	0.565		0.572	
Schwartz, 2003 A [20]	1.240		1.060	1.450	2.691		0.007	
Schwartz, 2003 B [20]	1.450		1.069	1.967	2.386		0.017	
	1.086		1.027	1.148	2.893		0.004	
**T2DM**	**Non-T2DM**		**95% CI**				**Statistic**	***p***
***n***	***n***	**Ages**	**OR**	**Lo**	**hi**	***p***		
4105	45,362	15–92						
*Effects Model*								
Fixed			1.09	1.03	1.15	0.004		
Random			1.22	1.04	1.44	0.02		
							Q = 49.51	0.0001
							I^2^ = 74.35	

^§^ Compiled in part from Wu et al. 2017. A, B—This study reported low (A) and high (B) cadmium exposure ORs separately.

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
