# Peer review of "Cadmium Is Associated with Type 2 Diabetes in a Superfund Site Lead Smelter Community in Dallas, Texas"

_ijerph, 2020, doi:10.3390/ijerph17124558_

Round 1

Reviewer 1 Report

Abstract

It is not clear the main hypothesis of the manuscript? What is the mechanisms  how the Cd-induced the T2DM?

After the classical conclusion, can the authors write any perspectives of the work?

Introduction

L.35-36. “It is…. metabolic processes”. Please, add a new reference that supports such affirmation.

L.43-49. Is there any relation to this paragraph with the the subject of the manuscript?

Please, re-write the introduction in the sense to point out the importance of cadmium.

L/ 51-53. It is presented here 9 references about Cd -T2DM. Which is the gap of this manuscript?

  1. Materials and Methods

L.60. 2.1. Study Population. How participants were selected for the study?The paragraph L61-73 does not fit to Material and Methods. 

  1. 75-89. The study from 2002 is the inspiration for this work. L.81 - All subjects in the present study were African American. Where do you start your study design? It is not clear.

After reading this part of the manuscript 2-3 times, I got that you are using the data of 2002. Is it correct?

Table 1. Please, run off the significant algharisms.

L127. Which is the consequence of an excess of females? Can this situation impact to the results?

L.116-146. The content presented is just a trivial translation from  table 1.

  1. 116-222. I think that would be more interesting to present the table 1 data and discuss it subsequently. It is hard for the reader to follow the discussion as it is.

L231-242. This is not part of the discussion. Move it to the introduction.

L243-257.  There is no reference to the discussion of the Possible Mechanism Pathway. Is there any other publication on the field to compare?

Discussion:

Authors should comment why they studied only Afro-Americans. Would the results be comparable in Caucasians?

Conclusion:

Which is the overall messages the readers take away?

Can you put the results into perspectives?

Author Response

Abstract

It is not clear the main hypothesis of the manuscript? 

To test the hypothesis that cadmium (Cd) exposure is associated with type 2 diabetes mellitus (T2DM).

What is ARE the mechanisms  how the Cd-induced the T2DM?

Available data suggest Cd exposure is associated with an increased propensity to increased insulin resistance.

After the classical conclusion, can the authors write any perspectives of the work?

Available data suggest Cd exposure is associated with an increased propensity to increased insulin resistance.

Introduction

L.35-36. “It is…. metabolic processes”. Please, add a new reference that supports such affirmation.

It is chemically similar to zinc, but unlike zinc, Cd is not part of normal animal metabolic processes. Its physicochemical properties allow it to assume positions in metabolic pathways that zinc and copper normally occupy [2, 3].

L.43-49. Is there any relation to this paragraph with the the subject of the manuscript?

It is important to understand the basic human exposure mechanisms of Cd, and we believe it is appropriately place in the paper.

Please, re-write the introduction in the sense to point out the importance of cadmium.

The Introduction has been re-written and references added accenting the importance of Cd.

L/ 51-53. It is presented here 9 references about Cd -T2DM. Which is the gap of this manuscript?   WHAT IS THE REVIEWER ASKING?

THIS REMARK IS VERY DIFFICULT TO DECIPHER, BUT I THINK THESE LAST LINES FROM THE REVISED MANUSCRIPT SPEAK TO THE POINT I THINK THE REVIEWER IS TRYING TO MAKE:

An estimated 12 studies have analyzed the association [10-21] and half of them [14-16, 18, 20] found a significant association of Cd with T2DM.   Analyses of large populations prospectively are needed to more thoroughly evaluate the association between Cd exposure and T2DM.  No clear consensus for the association was found across ten investigations [10-13, 15-20].

The purpose of our investigation is to analyze the association between T2DM and exposure to cadmium and several other heavy metals, and conduct a meta-analysis across published studies.

  1. Materials and Methods

L.60. 2.1. Study Population. How participants were selected for the study?

 The sample is not random.  It is a ‘sample of convenience,’ as frequently used in health surveys.  Of the 12 previously published studies used for comparison, only four were random samples.  Five studies were samples of convenience, two were NHANES secondary analyses (representative probability based cluster samples), and one was all workers in four coke factories in Wuhan, China.  The samples were all comprised mainly of working and lower SES individuals, very similar to the study sample. 

The paragraph L61-73 does not fit to Material and Methods. 

WE STRONGLY DISAGREE. THIS DESCRIBES THE STUDY POPULATION'S HISTORY AND HOW THIS COMMUNITY CAM TO BE EXPOSED TO CADMIUM. 

  1. 75-89. The study from 2002 is the inspiration for this work. L.81 - All subjects in the present study were African American. Where do you start your study design? It is not clear.

THIS QUESTION DOES NOT MAKE SENSE: Where do you start your study design?  WHAT DOES IT MEAN?

After reading this part of the manuscript 2-3 times, I got that you are using the data of 2002. Is it correct?

YES, THAT IS CORRECT, AND THE FIRST SENTENCE OF 2.2.

Table 1. Please, run off the significant algharisms.

"algharisms" IS NOT A WORD, AND I HAVE NO IDEA WHAT IT MEANS.

IF THE QUESTION IS TO ASK FOR US TO USE THE CHEMICAL SYMBOLS, THAT IS NOW DONE.

L127. Which is the consequence of an excess of females? Can this situation impact to the results?

"There was an excess of females in the T2DM group by 6% (p < 0.07), and 31.6% more of those aged >50 years in the diabetes group, which is similar to other studies of T2DM [17,20]."

THIS IS A COMMON FINDING IN T2DM STUDIES, DONE BY ME AND OTHERS IN THE US, MEXICO, AND POLAND.  THIS DOES NOT BIAS THE RESULTS BECAUSE FEMALES TEND TO HAVE A HIGHER PREVALENCE OF T2DM.  IN ADDITION, WOMEN PARTICIPATE IN STUDIES MORE FREQUENTLY THAN MALES BECAUSE MALES ARE OFTEN AT WORK WHEN SURVEYS ARE DONE.

L.116-146. The content presented is just a trivial translation from  table 1.

WE HAVE ADDED MORE STATISTICAL INFORMATION AND DISCUSSION TO THIS SECTION OF THE METHODS.

  1. 116-222. I think that would be more interesting to present the table 1 data and discuss it subsequently. It is hard for the reader to follow the discussion as it is.

WE HAVE EDITED THE INFORMATION DISCUSSING TABLE 1 AND ADDED BMI.

L231-242. This is not part of the discussion. Move it to the introduction.

WE HAVE MOVED THIS SECTION FROM DISCUSSION TO INTRODUCTION.

L243-257.  There is no reference to the discussion of the Possible Mechanism Pathway. Is there any other publication on the field to compare?

THIS IS DONE IN THE REVISED DISCUSSION; SUCH INFORMATION IS NOT TRADITIONALLY PRESENTED IN THE RESULTS SECTION. PATHWAYS ARE SOMETHING THAT WE HAVE NO DIRECT DATA ON.

Discussion:

Authors should comment why they studied only Afro-Americans. Would the results be comparable in Caucasians?

FROM SECTION 2.2 STUDY DESIGN

All subjects in the present study were African American 98% of the total sample); 2% of the sample had race missing, or were listed as Hispanic, and were excluded. The 2% were excluded to produce an ethnically homogeneous study population.

WE STUDIED AFRICAN AMERICANS BECAUSE THEY WERE 98% OF THE POPULATION. 

Conclusion:

Which is the overall messages the readers take away?

WE HAVE STATED THAT SMALL INCREASED RISK OF T2DM IS ASSOCIATED WITH CADMIUM EXPOSURE:

Can you put the results into perspectives?

LAST PARAGRAPH IN THE PAPER.

In conclusion, Cd apparently increases risk for T2DM, meta-analysis of ten studies combined supports the increased risk, and the association is biologically plausible. Collectively, available data support a small contribution of Cd to T2DM risk, between 9% and 22%. Prior studies that did not find a significant association of Cd exposure with had T2DM may have suffered from a Type II statistical error.

Reviewer 2 Report

This study from Little et al. conducted a cross-sectional survey on the association between Cadmium exposure and Type 2 Diabetes in a Superfund Site Lead Smelter Community in Dallas, Texas. This study was generally well written and presented. I have some suggestions for the authors’ considerations before I can accept this paper for publication.

Major comments:

  1. Introduction should provide more sufficient background and include all relevant references.
  2. It says in lines 76-79 “Participants were 76 recruited through town hall meetings and public service announcements in the media (newspapers, 77 radio, and television). People who lived in the following zip codes were included: 75208, 75211, 75212, 78 75247, 75203, 75215, and 75216. The vast majority were from the West Dallas smelter community.” The authors should further clarify and provide rationality how can the participants included in the survey represent the population, it seems that the participants were not a random sample.
  3. In the analytic models, the authors failed to include the covariate of BMI. I think it’s more appropriate to include BMI as an ordinal rather than continuous variable in the logistic models. More descriptions about this should be included.
  4. Are there any potential collinearity problems when simultaneously including log10 (blood lead level), log10 (blood arsenic level), log10 (blood mercury level), and log10 (blood cadmium level) in the analytic models? The authors should be cautious with this. Maybe the authors can perform a number of sensitivity analyses.

Minor comments:

  1. Lines 15-16: “A two-phase health screening (physical 15 examination and laboratory tests) conducted....” should be “...was conducted...”.
  2. Lines 22-23: P-value in this statement “T2DM individuals 22 (70.3%) were >50 years old (p < 0.00001)” was unclear.
  3. Some modifications should be made for the statement (lines 26-28) “In meta-analysis of 12 prior studies and 26 this one, T2DM risk was OR = 1.09 (95% CI: 1.03–1.14, p < 0.004) fixed effects and 1.22 (95% CI: 1.04–27 1.44, p < 0.02)”, maybe “In meta-analysis of 12 prior studies and this one, T2DM risk was 1.09 (95% CI: 1.03–1.14, p < 0.004) in fixed-effect model and 1.22 (95% CI: 1.04–27 1.44, p < 0.02) in random-effect model, respectively, associated with a 1 μg/L increase in Cd exposure.....”.
  4. “Multiple logistic regression” in line 96 should be “Multiple-variate logistic regression”.
  5. More explanations should be included to clarify how the associations between Cadmium exposure and Type 2 Diabetes were expressed or interpreted using odds ratio. Are the ORs were associated with a 1 μg/L increase in Cd exposure???

Author Response

Major comments:

  1. Introduction should provide more sufficient background and include all relevant references.

WE HAVE ADDED SEVEN REFERENCES AND EXPANDED THE DISCUSSION: 

Cadmium (Cd) is a trace element, occurring in about 0.1 to 0.5 ppm in the earth’s crust [1]. It is chemically similar to zinc, but unlike zinc, Cd is not part of normal animal metabolic processes. Its physicochemical properties allow it to assume positions in metabolic pathways that zinc and copper normally occupy [2, 3]. Cd is toxic to higher organisms because it disrupts normal metabolism and accumulates in organs. Cd accumulates in kidney in humans, and is particularly nephrotoxic because it concentrates in proximal tubular cells [4]. Secondarily, Cd accumulates in lungs, bone and liver [5]. In addition, Cd is associated with type 2 diabetes mellitus (T2DM) onset, cardiovascular disease and osteoporosis [2, 6].

4.1. Cadmium Toxicity Profile in Humans

Cd is toxic to humans, and was first recognized in 1858 [7]. Uptake through the gastrointestinal route is small, approximately 5% of ingested Cd, although Cd can accumulate in vegetable foods. Human lung resorbs 40–60% of Cd in tobacco smoke, which is the major source of Cd inhalation [1]. Once absorbed, the majority of Cd circulating in blood is bound to proteins (e.g., albumin, metallothionein). Liver is the first organ to process absorbed Cd, and where Cd induces the production of Cd-metallothionein [7]. Kidney damage associated with Cd exposure is a well-recognized problem for exposed patients [2]. Cd reaches the kidney in the form of cadmium-metallothionein (Cd-MT). Cd-MT is filtered by the glomerulus, and subsequently reabsorbed in proximal tubule, where it remains, comprising the major part of Cd body burden. Cd concentration in proximal tubular cells increases over a person's life span, with toxicity expressed as chronic kidney disease. In this study, the frequency of CKD3 was increased in frequency among those exposed to Cd [8].

Cd exposure in humans is through inhalational (tobacco smoking) and oral (food) routes, but absorption from inhalation accounts for the greatest accumulation. Cd accumulates in the mass of most plants, particularly tobacco plants. Smokers consequently have kidney Cd concentrations 4 to 5-fold greater than the general non-smoking population, and these concentrations persist for decades. For non-smokers, the main source of Cd exposure is food consumed. Cd is concentrated in cereals (i.e., cereal products), vegetables, nuts and pulses, starchy roots or potatoes. In turn, those animals fed these products subsequently concentrate Cd in meat and meat products. Cd exposure also occurs environmentally (e.g., dust, airborne particulates) from smelting activities (e.g., lead, zinc, copper).

T2DM is associated with Cd exposure as measured in blood/urine levels, often from exposure to by-products of lead smelting. Published literature has reported that Cd exposure causes increased insulin resistance [9]. Increased insulin resistance ultimately increases the propensity to develop T2DM. Cd increases pro-inflammatory action and monocyte chemoattractant protein-1 (MCP-1) expression is upregulated. Adipose tissue Cd accumulation leads to increased endocrine dysfunction and increased lipolysis. Adipocyte glucose dysregulation contributes to hyperglycemia. These and other influences increase the propensity to have insulin resistance [9]. An estimated 12 studies have analyzed the association [10-21] and half of them [14-16, 18, 20] found a significant association of Cd with T2DM.   Analyses of large populations prospectively are needed to more thoroughly evaluate the association between Cd exposure and T2DM.  No clear consensus for the association was found across ten investigations [10-13, 15-20].

The purpose of our investigation is to analyze the association between T2DM and exposure to cadmium and several other heavy metals, and conduct a meta-analysis across published studies.

2. It says in lines 76-79 “Participants were 76 recruited through town hall meetings and public service announcements in the media (newspapers, 77 radio, and television). People who lived in the following zip codes were included: 75208, 75211, 75212, 78 75247, 75203, 75215, and 75216. The vast majority were from the West Dallas smelter community.” The authors should further clarify and provide rationality how can the participants included in the survey represent the population, it seems that the participants were not a random sample.

The sample is not random.  It is a ‘sample of convenience,’ as frequently used in health surveys.  Of the 12 previously published studies used for comparison, only four were random samples.  Five studies were samples of convenience, two were NHANES secondary analyses (representative probability based cluster samples), and one was all workers in four coke factories in Wuhan, China.  The samples were all comprised mainly of working and lower SES individuals, very similar to the study sample. 

3. In the analytic models, the authors failed to include the covariate of BMI. I think it’s more appropriate to include BMI as an ordinal rather than continuous variable in the logistic models. More descriptions about this should be included.

WE HAVE INCLUDED BMI IN THE DESCRIPTIVE STATISTICS. IT WAS IN THE LOGISTIC REGRESSION. WE ALSO HAVE A RATIONALE FOR USING THE SCALED BMI INSTEAD OF THE CATGORIZED BMI: 

BMI was used as a continuous variable because each point change in the BMI is meaningful at the high end of the BMI distribution, and it has a standard error of 1. We tested using categories of overweight and obesity, and found that all subjects were obese (BMI > 30).  

  1. Are there any potential collinearity problems when simultaneously including log10 (blood lead level), log10 (blood arsenic level), log10 (blood mercury level), and log10 (blood cadmium level) in the analytic models? The authors should be cautious with this. Maybe the authors can perform a number of sensitivity analyses.

GOOD QUESTION.  WE NOW PRESENT THE VARIANCE INFLATION FACTOR (VIF) AND THE ANALYSES SHOW ALL VARIABLES HAD A LOW VIF. 

Multicollinearity was assessed with multiple regression variance inflation factor (VIF) using the model above. VIFs were generally, and were less than 1.45.  The VIFs for demographics (age – 1.41, gender – 1.27, smelter worker – 1.19, duration – 1.10, smoker – 1.09), physical characteristics (BMI – 1.17, hypertension -- 1.27, GGT – 1.13), and heavy metals (Ar – 1.08, Cd – 1.17, Hg – 1.08, Pb – 1.29). These VIFs indicate multicollinearity is not confounding these results.

Minor comments:

  1. Lines 15-16: “A two-phase health screening (physical 15 examination and laboratory tests) conducted....” should be “...was conducted...”. CORRECTED
  2. Lines 22-23: P-value in this statement “T2DM individuals 22 (70.3%) were >50 years old (p < 0.00001)” was unclear. CORRECTED.
  3. Some modifications should be made for the statement (lines 26-28) “In meta-analysis of 12 prior studies and 26 this one, T2DM risk was OR = 1.09 (95% CI: 1.03–1.14, p < 0.004) fixed effects and 1.22 (95% CI: 1.04–27 1.44, p < 0.02)”, maybe “In meta-analysis of 12 prior studies and this one, T2DM risk was 1.09 (95% CI: 1.03–1.14, p < 0.004) in fixed-effect model and 1.22 (95% CI: 1.04–27 1.44, p < 0.02) in random-effect model, respectively, associated with a 1 μg/L increase in Cd exposure.....”. RE-WORDED ACCORDINGLY.
  4. “Multiple logistic regression” in line 96 should be “Multiple-variate logistic regression”. WE CHANGE THE LABEL TO MULTIVARIATE LOGISTIC REGRESSION.
  5. More explanations should be included to clarify how the associations between Cadmium exposure and Type 2 Diabetes were expressed or interpreted using odds ratio. Are the ORs were associated with a 1 μg/L increase in Cd exposure???

GOOD QUESTION. WE MADE AN ERROR. IT SHOULD BE A UNIT INCREASE IS ACTUALLY 10 μg/L.  THANK YOU.

Reviewer 3 Report

The authors report an analysis of cadmium exposure and other potential explanatory factors on type 2 diabetes risk in communities affected by smelter operations.  I am concerned that the association reported results from bias due to the self-selected population that was studied.  This is a significant challenge and at a minimum the authors must consider this in their overall analysis and discussion of the findings.  It may be informative to more carefully explore the studies collected for their meta-analysis.  How does their study population compare with those of other studies in the meta-analysis?  Are there differences in the populations studied that explain differences in study findings? 

The study does have strengths - in objective assessment of exposures and outcomes - but all these strengths and limitations must be fully disclosed and discussed before the scientific soundness can be determined.

Background on cadmium toxicity and potential mechanisms related to diabetes are included at the end of the paper (sections 4.1-4.2).  These sections should be placed in the Introduction as part of the scientific rationale for the study.

Methods are not fully described: The authors must explain how they conducted their literature search to find the articles included in the meta-analysis.  It should be a systematic review.

Editorial changes needed:

Lines 36-42: references are needed in this section

Line 71: typo: Environmental Protection Agency

Lines 103-109: better explanation of literature search

116-117: There is an incomplete sentence here: "More individuals smaller CIs"

159-161: These are repetitive sentences - one can be deleted.

164: Incorrect statement.  BMI is significant

192-194: Need to re-word this sentence

Author Response

(The authors gave the same response as above.)

Round 2

Reviewer 2 Report

accepted

Author Response

No comments to address. thanks.

Reviewer 3 Report

The revisions done to date have improved the paper to a point.  Prior to publication the methods section must be improved and several other corrections are needed as listed below.

Section 2.4 Analytical techniques (Methods section)

At a minimum this section must have 2 paragraphs; one to explain the analysis of the survey data and the other to clearly describe in further detail the literature review.  

Lines 120-121 - introduce both analyses: 1) survey data analysis; and 2) literature review

Lines 133-134 - Start a new paragraph here to describe the literature review.  It would be preferable for the literature search to include both PubMed and Embase (to capture European literature).  However, at a minimum further description of the PubMed search including the date range of the search is needed, for example, the search captured articles published between the dates of January 1, 1990 and December 31, 2019.  The authors should also include a sentence to describe the nature of articles that were excluded.

Line 135: correct grammatical error, "searched"

Other corrections:

Line 149-150: Remove or correct the incomplete sentence here "More individuals smaller CIs"

Line 199: BMI is shown as a significant predictor in Table 2 for >50 year olds

Line 228: Reword the awkward sentence here, "The present primary findings are higher than those in the done after..."  

Author Response

Section 2.4 Analytical techniques (Methods section)

At a minimum this section must have 2 paragraphs; one to explain the analysis of the survey data and the other to clearly describe in further detail the literature review.  

A PRISMA compliant literature review was done. Using pubmed.gov and scholar.google.com for years 1960 to December 31, 2019, search terms were (Cadmium AND Diabetes), which resulted in 372 studies. A subset of those studies was searched using (Human AND Type 2 Diabetes) to refine the search to humans with T2DM, with 32 studies meeting those search criteria. Manual review excluded studies were those that did not report primary data, (described below), or did not report Cd levels or method of testing.  Presence of primary data on study parameters was required for inclusion of studies, as described below, resulted in 12 studies of Cd and T2DM for comparison with our results, excluding studies that reported only secondary data. Studies chosen for inclusion had to present primary data analysis on the association of Cd with T2DM. Investigations included were limited to those that reported sample sizes of Cd exposed and controls, and those who had T2DM. In addition, the chemical method of determination had to be included, and the biological fluid used for analysis (blood, urine), and T2DM diagnostic criteria.

Lines 120-121 - introduce both analyses: 1) survey data analysis; and 2) literature review

Two analytical approaches were used in the present study. We used logistic regression to analyze the ability of demographics and heavy metals to predict T2DM status. Ana analysis of the published literature was done using a systematic literature search and meta-analysis of studies that met inclusion criteria (reported primary data, sample sizes, Cd levels, method of Cd level determination).

Lines 133-134 - Start a new paragraph here to describe the literature review.  It would be preferable for the literature search to include both PubMed and Embase (to capture European literature).  However, at a minimum further description of the PubMed search including the date range of the search is needed, for example, the search captured articles published between the dates of January 1, 1990 and December 31, 2019.  The authors should also include a sentence to describe the nature of articles that were excluded.

We did not use Embase; Google Scholar captured European publications that Pubmed.gov did not.

Meta-analysis was done to obtain estimated composite ORs and 95% Confidence Intervals (CI) under fixed and random effects, and measures of heterogeneity. The heterogeneity was assessed using the Q and I2 statistics. A PRISMA compliant literature review was done. Using pubmed.gov and scholar.google.com for years 1960 to December 31, 2019, search terms were (Cadmium AND Diabetes), which resulted in 372 studies. A subset of those studies was searched using (Human AND Type 2 Diabetes) to refine the search to humans with T2DM, with 32 studies meeting those search criteria. Manual review excluded studies were those that did not report primary data (described below), or did not report Cd levels or method of testing.  Presence of primary data on study parameters was required for inclusion of studies, as described below, resulted in 12 studies of Cd and T2DM for comparison with our results, excluding studies that reported only secondary data. Studies chosen for inclusion had to present primary data analysis on the association of Cd with T2DM. Investigations included were limited to those that reported sample sizes of Cd exposed and controls, and those that reported T2DM status. In addition, the chemical method of determination had to be included, and the biological fluid used for analysis (blood, urine), and T2DM diagnostic criteria.

Line 135: correct grammatical error, "searched"

Done.

Other corrections:

Line 149-150: Remove or correct the incomplete sentence here "More individuals smaller CIs"

Done.

Line 199: BMI is shown as a significant predictor in Table 2 for >50 year olds

Corrected.

Line 228: Reword the awkward sentence here, "The present primary findings are higher than those in the done after..."  

Corrected.